# Ecosystem Services: A Social and Semantic Network Analysis of Public Opinion on Twitter

**DOI:** 10.3390/ijerph192215012

**Published:** 2022-11-15

**Authors:** Stefano Bruzzese, Wasim Ahmed, Simone Blanc, Filippo Brun

**Affiliations:** 1Department of Agricultural, Forest and Food Sciences (DISAFA), University of Torino, Largo Paolo Braccini 2, 10095 Grugliasco, Italy; 2Management School, University of Stirling, Stirling FK9 4LA, UK

**Keywords:** ecosystem services, social network analysis, content analysis, semantic analysis, NodeXL

## Abstract

Social media data reveal patterns of knowledge, attitudes, and behaviours of users on a range of topics. This study analysed 4398 tweets gathered between 17 January 2022 and 3 February 2022 related to ecosystem services, using the keyword and hashtag “ecosystem services”. The Microsoft Excel plugin, NodeXL was used for social and semantic network analysis. The results reveal a loosely dense network in which information is conveyed slowly, with homogeneous, medium-sized subgroups typical of the community cluster structure. Citizens, NGOs, and governmental administrations emerged as the main gatekeepers of information in the network. Various semantic themes emerged such as the protection of natural capital for the sustainable production of ecosystem services; nature-based solutions to protect human structures and wellbeing against natural hazards; socio-ecological systems as the interaction between human beings and the environment; focus on specific services such as the storage of atmospheric CO_2_ and the provision of food. In conclusion, the perception of social users of the role of ecosystem services can help policymakers and forest managers to outline and implement efficient forest management strategies and plans.

## 1. Introduction

### 1.1. Ecosystem Services

Ecosystems provide benefits to humans, and these services are known as ecosystem services [1]. They are, for example, forest purification of surface water [2], carbon sequestration [3,4] and psycho-physical well-being from frequenting green spaces [5]. These services underpin the achievement of some of the Sustainable Development Goals defined in the U.N. 2030 Agenda [6]. There are several classifications of such services in the literature, the first of which is provided below [1]. Ecosystem services can be divided into: “supporting services” that have an indirect effect on society but serve for the creation of all other categories of services; these are, for example, nutrient cycling and soil formation [7]; “provisioning services” that provide a tangible ecosystem benefit to society, such as drinking water and firewood [8]; “regulating services” that results from the proper management of ecological processes within an ecosystem, such as the reduction in natural disasters, the purification of surface water and the mitigation of ongoing climate change [9]; “cultural services” that provide intangible benefits, such as spiritual value, the aesthetic beauty of a landscape and recreational activities [10,11].

Ecosystem services now find an increasing interest in civil society [12] and also in the digital domain, as evidenced by the average search frequency of the term “ecosystem services” in the last year globally on the Google search engine, which is about 64% [13]. However, only a little research has addressed the public perception of such services, focusing instead mainly on technical and economic aspects [14]. A better understanding of such perception would help in forest planning processes, green space design and natural capital protection and enhancement [15].

### 1.2. Social Media Data

Social media can be useful for extracting information and insights into subjective perceptions of a particular topic [16] and finding various uses in the academic context [17,18]. According to the Digital 2022 Global Overview Report [19], there are over 4.6 billion users on social media, three times as many as a decade ago and 10% more than in 2021. This figure is growing exponentially every year, at a faster growth rate than with the Internet, and translates into a huge amount of data at our disposal [20].

Among the different social media present nowadays, Twitter has been highly successful in academic research, as it is the main social platform where scientists and researchers disseminate their work and raise awareness [21,22], although it ranks 14th in the ranking of the most active social media platforms in the world [23]. In 2017, 1–5% of Twitter users were still active scientists [24,25], a number that is still growing today [26]. Another important aspect is that on Twitter, unlike other social media, fake news and disbeliefs find it difficult to spread [27].

### 1.3. Literature Review

Several studies in the literature have analysed ecosystem services using social media [28,29,30]. Most authors, however, have focused exclusively on cultural ecosystem services by mapping and quantifying them using images [31,32,33,34], and few of them have analysed their social perceptions [15,35,36]. With reference to Twitter, several authors have analysed the social perception of natural capital, that is, nature and its ecosystem services and protected areas [37,38,39] or people’s emotions [40], but few have focused exclusively on ecosystem services and specifically cultural services [41,42,43,44]. The only study that analysed the perception of ecosystem services in a broad sense focused on a specific geographical area, namely the Laurentian Great Lakes [45].

### 1.4. Research Questions and Hypothesis

In light of the findings of the literature review, this study aims to investigate and fill research gaps regarding the global-scale social perception of ecosystem services on social media. Specifically, the research was conducted on Twitter with the aim of answering the following questions:(RQ1) What kind of social network and what characteristics do we find on Twitter related to ecosystem services?(RQ2) Which users play a key role in the dissemination/disruption of information on the network?(RQ3) What are the most discussed topics and keywords that emerge in the discussions?

Based on the empirical evidence found in the literature [38,39], the topic under research and our theories, the respective hypotheses for the research questions were identified:(H1) In the social network, there are good interactions between users and high content sharing.(H2) Most influential users in the network are scientists or researchers.(H3) The most discussed topics are related to the latest research in the literature on ecosystem services, while the keywords are related to both technical terms used in research and more generic terms known to society.

### 1.5. Sections

The manuscript presents a first section devoted to a brief description of the subject matter and the context of the research. Section 2 describes the theoretical foundations on which the social and semantic network analyses are based, and the methodological flow adopted. Section 3 and Section 4 report and comment on the results obtained. The last section concludes with the limitations of this study and its possible developments.

## 2. Materials and Methods

### 2.1. Data Collection

This study gathered 4398 tweets that were sent between 17 January 2022 and 3 February 2022, using the keyword of “ecosystem services” and the hashtag “ecosystemservices”. The Microsoft Excel plugin, NodeXL [46] was used to retrieve data. The data collection was conducted in two phases. The first phase was to retrieve data on 25 January, building the first dataset that had a time range of 17 to 25 January and a second dataset was captured on 3 February 2022, that had a time range of 26 January to 3 February NodeXL has access to Twitter’s Search Application Programming Interface (API), which provides official access to data. The Search API provides data going back in time around seven days. For low-tweet volume topics, as in our case, the Search API was able to retrieve all the available tweets sent during our time period of data collection.

### 2.2. Data Analysis

The retrieved data was analysed in NodeXL by drawing upon social network and semantic network analysis (Appendix A), and the social network analysis visuals were produced in the Gephi software application. Social network analysis (SNA) is the examination of social structures using networks and graph theory. It characterises networked structures in terms of nodes or entities that are connected by ties or relationships between the entities. In sociology, it is often used to study social groups and has also been utilised in management consultancy to study relationships among employees. In the context of Twitter, SNA can shed light on how users connect with one another.

More specifically, by using SNA and analysing how users interact with each other (retweet, reply, mention) it is possible to build an understanding of the most popular users and groups of users. SNA methods also help to visualize the users and interaction patterns in social networks. In this study, we provide a visual representation of the interaction of users and are able to show the most popular groups (communities with the most users) and the most influential users within the network (weighted by size).

Newcomers to SNA may want to consult an overview and classification of social media networks published by Himelboim et al. [47]. The network graph was produced by analysing relationships between Twitter accounts. Accounts that conversed with each other were assigned into distinct groups within the network. Influential users were identified by using a centrality algorithm called betweenness centrality, which ranks users according to the number of shortest paths that pass through them [48]. Figure 1 provides a visual summary of this study’s methodology.

## 3. Results

### 3.1. Search Results

Figure 2 shows the different types of tweets published in the two time periods analysed. An initial comparison of the two datasets reveals an almost similar situation between the different types of tweets published. The first dataset collected 2317 different types of tweets in total, while the second 2081 over a data collection time of 7 days backwards in time. This first output allows us to make a consideration, namely a low interest in the term ecosystem services on Twitter during the analysed period, since out of 18,000 types of tweets, which was the maximum threshold set in the analysis, only 4398 were collected. The ratio between tweets and retweets, shifted more towards the latter, suggests instead an excessive noise of information circulating in the network, as the number of shared contents is higher than new contents.

### 3.2. Social Network Analysis

#### 3.2.1. Network Overviews

Table 1 shows the macroscopic characteristics of the social network. Relationships are the connection between two Twitter users. We speak of relationships with duplicates in the case where there are multiple connections between two users. In both analysed datasets, the number of relationships with duplicates is very low, which probably suggests that users merely share content but do not engage in discourse or debate about it, and this is also confirmed by the reciprocated Twitter users pair ratio, as only 2 out of 100 users are mutually connected. The number of isolated Twitter users, namely those who have no relationship with other users, is low for both datasets. The diameter shows the size of the social network is, in our case, both diameters are high, so the speed of information dissemination is slow within the network. However, the low average shortest path suggests the opposite, namely that there are influential users in the network, referred to as “gatekeepers”, which speed up the information dissemination process. Finally, the density and modularity metrics, both of which have a value range from 0 to 1, explain the structure of the network. Specifically, the density of both networks is very low, suggesting that most users have no relationship with others. Modularity, on the other hand, is very high, thus, there are well-structured user groups within the networks that are strongly intra-connected but weakly interconnected with other groups. In summary, information circulates slowly in the entire network but quickly in the groups due to the presence of gatekeepers.

#### 3.2.2. User Analysis

Table 2 and Table 3 show the top ten most influential users across the two datasets. The first aspect that emerges is that the users between the two networks are different, as the analyses are made about a week apart. The stakeholders involved are multiple, such as civil society, scientific journals, non-profit organisations, government initiatives and, to a greater extent, academia with professors, scientists and scholars related to environmental issues. Most influential users belong to a few groups that are recurrent in social networks, which probably indicates that these groups have the largest size. The important metric for a user’s influence is betweenness centrality, since the higher it is the more influence the user has in allowing or blocking the exchange of information between one person/user and another in the network. The number of followers, instead, is not an important metric for determining the influence of a user. High follower numbers may be there because users pay for fake followers or bots. It was intended to include this metric to show how there is no causal link between a user’s influence and his or her number of followers. In Table 2, the most influential users are part of the academy, suggesting a research interest in the topic of ecosystem services.

In Table 3, the most influential users are a global network related to ecosystem services and a trust fund dedicated to environmental threats, most likely because of the anniversary of World Wetlands Day, promoted annually on 2 February by the United Nations.

### 3.3. Content Analysis

#### 3.3.1. Top Contents

Hashtags in Twitter serve to categorise topics and allow users to follow them more easily. Table 4 shows the top ten hashtags of the two datasets. What emerges, excluding the search term, is that some popular keywords today are common to both social networks, such as “ecosystem”, “biodiversity” and “climatechange”. There is also the presence of specific hashtags linked to events or anniversaries, such as “india” and “tiger”, in reference to the news about the increase in tiger reserves in India, which as protected habitats indirectly protect ecosystem services, or “worldwetlandsday”, “wetlands”, “actforwetlands” and “worldwetlandday” linked to raising awareness about the importance of wetlands and the services they provide to society and the environment. The hashtag “groundedinsoil” is used by the Canadian Society of Soil Science to inform about the ecosystem services provided by soil, and probably the keyword “soil” was matched, as it has the same occurrence.

#### 3.3.2. Semantic Analysis

Table 5 shows the total number of word-pairs and their frequency classes for the two time periods analysed. In the first dataset, for example, 4053 word-pairs appeared from 0 to 2 times, 1711 from 3 to 10 and so on. The more frequent a word-pair is, the more important it is. The reverse trend is clear, namely, as the frequency increases, the word pairs decrease and vice versa. No major differences emerge between the two datasets in terms of the number of word-pairs per frequency. The most frequent word-pair obviously refers to the search term “ecosystem services”. For the semantic network analyses, only word-pairs with a frequency of at least ten times were considered to avoid too much noise in subsequent processing.

Table 6 shows the most frequent word pairs in the analysed tweets. Due to space constraints, only the top ten results are shown for both datasets. Knowing the word-pairs helps to identify the topics discussed within social networks [49]. In general, the results that emerge are in line with the type of influential users within the two networks, since in the first case, we talk about human–nature interactions and social-ecological systems, which are much-studied topics in academia today, and in the second case, on the other hand, aspects related to climate change, biodiversity and policies emerge that are often discussed or addressed by non-governmental and non-profit organisations, activists and trust funds with the aim of raising awareness of the civil society with respect to these issues. Several word pairs then confirm what has already been identified with hashtags, namely the event in India on the increase in tiger reserves and World Wetlands Day 2022.

### 3.4. Network Visualisations

#### 3.4.1. Social Network Analysis

Figure 3 below is a network visualisation of the dataset retrieved on 25 January 2022. 

The users’ most influential appear towards the centre of the network, and their size highlights their influential nature. The network visual highlights how discussions occur in different groups and clusters. The different colours represent the different groups conversing in the network, with a few very influential users driving some larger discussions. The most dominant cluster is coloured in purple and makes up 7.29% of the network. This is then followed by clusters in green (5.89%) and blue (4.7%). The network can be characterised as a community of users who connect over a shared interest.

Figure 4 below is a network visualisation of the dataset retrieved on 3 February 2022.

The network retrieved on 3 February 2022 appeared similar in structure to the network in January. There appear to be several key groups and only a handful of users who are particularly influential in shaping the discussion. The most dominant cluster was coloured purple, which made up 7.43% of the network, followed by green (7.28%) and blue (6.04%). It is worth mentioning the dark grey cluster (5.3%) of the network because it contained the user with the highest betweenness centrality score, which is a measure of network influence.

In both network visualisations, there are multiple smaller communities and conversations taking place with a few larger discussions. There are several medium-sized communities but no large or very large clusters of users. This finding, alongside the volume of tweets, indicates that it is a niche topic as popular topics tend to be much larger in volume and contain larger size communities conversing with one another. These results can be interpreted alongside Table 2 and Table 3, which provide insight into the specific users who were influential. 

#### 3.4.2. Semantic Networks

Different topics are discussed within the two networks analysed. In some cases, it is also possible to read pieces of sentences, pairing the different words that emerged from the semantic analysis. In Figure 5 and Figure 6, several topics related to specific aspects of ecosystem services emerge, each discussed in a different group. Some of these main topics are then recurrent in both social networks, such as sustainable food production, protection service against natural hazards and disasters (nature-based solutions), and regulation service related to carbon sequestration. Other topics, however, are unique to the networks, such as social-ecological interactions, support services and natural capital and the event on the expansion of tiger reserves in India in Figure 5 (25 January); the ecosystem services value chain and its markets, cultural services related to tourism and Wetlands Day in Figure 6 (3 February).

## 4. Discussion

Social media, as argued by Wang et al. [50], are a data source for analysing the behavioural patterns of individuals and for a better understanding of public opinion about specific issues. In this regard, several studies have made use of social media for applications on ecosystem services. Wang et al. [51] have performed research on the prioritisation of ecosystem services for cost-efficient governance. Xuezhu and Lange [14] explored the public perception of ecosystem services for nature-based solution projects. Hausmann et al. [52] analysed ’tourists’ preferences in protected areas for nature-based experiences. Other authors have focused on the assessment and mapping of such services [41,53,54].

However, there are no studies in the literature that have been carried out exclusively to understand the social network and public opinion on social media regarding the term ecosystem services. Yet, it is an increasingly popular topic used to indicate the public role of natural resources when moving from multifunctionality to the perceived usefulness of resources for society. This is the context of our research, and the choice of using Twitter as a case study is due to multiple reasons, as reported by several authors [55,56,57,58]. These include the ability to have open communication between users and therefore favour the sharing of opinions, compared to other private social networks, the increasing use of Twitter for research purposes and public opinion studies.

Based on these considerations, our work tried to shed light on the following questions:

1. The first one concerned the type and characteristics of the network within the term ecosystem services.

Based on the density and modularity metrics and the number of isolated Twitter users, it can be concluded that the structure of our network is “community clusters”, as defined by Smith et al. [59]. This result partially satisfies our hypothesis since the “community cluster” structure, as evidenced by the low density and number of isolated users and high modularity, restricts the flow of information to small groups of users who, in turn, rely on one or a few influential users in that cluster and for that specific topic related to ecosystem services. Himelboim et al. [47], however, identify advantages of this structure such as the stability of the small groups over time and the possibility of identifying different opinions in the network regarding the analysed topic since there is no single source of information and multiple conversations can be established each with its own community and points of view.

Another demonstration is the low relationship and reciprocal relationships in the network. Even though users made efforts to create relationships with other users, as evidenced by the high retweeting, the interactions were unsuccessful. Ahn and Park [60] state that the function of retweeting news is not only to facilitate the exchange of information but also to create relationships with other users and express their agreement in public. However, according to Wiiava and Handoko [61], a conversation is active when community users interact with each other.

This network structure, however, is in line with the study conducted by Pan and Vira [39] on natural capital, a term akin to ours, and differs from another environmental issue, namely climate change, in the study conducted by Williams et al. [62] on climate change, perhaps because the latter issue creates a strong polarisation and alignment in this regard. Although, a more recent study by Bridge [63] on climate action showed that the debate was not so polarised. The type of network probably changes depending on the time period, the topic analysed, and the importance and role of the users participating in the debate.

2. The second research question was about the type of users who play a key role in the information flow of the network. Here again, our hypothesis was partially confirmed, as in the social network obtained from the first dataset, the majority of the ten most influential users belonged to the academic world and published posts related to the promotion and approval of their research work, thus supporting what Côté e Darling [21] defined that Twitter is the main social platform for the dissemination of results from research.

The second network, on the other hand, had a majority of governmental bodies, non-profit organisations and activists, probably because in that analysed time period, there was World Wetlands Day, and several institutions were therefore intent on raising awareness on the topic. Aguilar-Gallegos et al. [64], in a study analysing the social networks on Twitter regarding an agricultural research and training centre in Mexico, point out that central institutions should not only create content but also identify and promote relevant information in the networks, act as intermediaries and encourage the dissemination of content. This concept is shared by several authors [65,66,67].

3. The last research question investigated the most discussed topics on the topic of ecosystem services. In order to do this, the analysis of the ten most used hashtags in tweets was useful, as it made it possible to identify topics of general interest, as stated by Fedushko [68] and remove disambiguation between tweets that do not have a commonality and obtain information about them, as reported by Palos-Sanchez et al. [69]. The main hashtags were the forerunners of semantic network analysis, and topics emerged that were both topical in research, such as social-ecological systems, nature-based solutions and markets for ecosystem services, and topics related to events or anniversaries. This diversity of topics affirms what was said before, namely that in “community clusters” networks, there are different sources of information, each with its own audience and conversations.

## 5. Conclusions

This study analysed the most discussed topics around ecosystem services on a social media platform, specifically Twitter. To our knowledge, this is the first study to use SNA and semantic analysis to analyse the social network and public opinion of the term ecosystem services on Twitter. Furthermore, this is one of the first studies using this methodology applied to ecosystem services in the post-pandemic period. What emerges is that the social network is characterised by having small, well-focused discussion groups on precise aspects related to ecosystem services, but that the flow of information throughout the network is weak. There are several influential users within the groups, and these vary depending on the time analysed and the topics discussed. The topics are also diverse and cover ecological, social, economic, and political aspects related to ecosystem services.

This study aimed to investigate and fill research gaps regarding social perceptions of ecosystem services on a global scale on social media, with the objectives of: characterising the social network on Twitter in relation to ecosystem services; identifying users who play a key role in the dissemination/dissemination of information on the network; and highlighting the most discussed topics and keywords that emerge in discussions.

The user analysis showed that there are many stakeholders involved, such as civil society, scientific journals, non-profit organisations, government initiatives and, to a greater extent, academia with professors, scientists and scholars related to environmental issues. Within this network, the most influential users are part of academia, suggesting the growing research interest in the topic of ecosystem services.

Content analysis summarises today’s most popular keywords in discussions, such as “ecosystem”, “biodiversity” and “climate change”. Semantic analysis confirms these results, and the most frequent word pair obviously refers to the search term “ecosystem services”.

### 5.1. Limitations

There are five potential limitations to the study. The first concerns the use of social media for research purposes, as such analyses are often difficult to replicate due to data access constraints and the need to rely on private companies specialising in data storage. The second relates to the choice of using only Twitter and its inherent biases, as each social media outlet has specific types of users and content posted, and this can influence the research. In any case, for our public opinion research, Twitter was the most suitable social platform, compared to others such as Instagram, which is mainly used to share recreational activities and experiences, Flickr to publish professional photos or Facebook, which with private and semi-private profiles makes it difficult to have open communication. The third limitation concerns the period considered, as our research only focused on two weeks, but it might be interesting to analyse more distant and relevant periods. The fourth limitation is due to the difficulty, in recent years, of analysing public opinion on ecosystem services in normal times, i.e., without the presence of special events or emergencies, such as political issues and socio-economic aspects, which can influence opinion and spark extraordinary debates and discussions. For this reason, in this work, an attempt was made to analyse two different time periods, to try to extract both the spot event component (period-specific) and the common component typical of normal times (inter-period). The last limitation is due to the choice of the English language only, but we only wanted to focus on the international level, and taking into consideration other languages would have drastically increased the number of tweets to be analysed by the software, with its consequent technical limitation of data processing. In fact, beyond the threshold of 18,000 tweets, the software is not able to collect any more data.

### 5.2. Implications for Research

This study can be used as a bibliographic source for future studies on public opinion of ecosystem services, as it is the first of its kind after the pandemic period. At the same time, it can be cited for the methodology applied, that is, SNA and semantic analysis on innovative tools such as social media. The results obtained can then be used in a combined approach with more traditional public opinion survey methods, such as interviews or questionnaires. Furthermore, our study builds upon previous literature using social network tools to analyse Twitter data [70,71,72]. Researchers can use the results of this study to conduct new analyses on ecosystem services by extending the research time period and refining the methodology employed. 

### 5.3. Implications for Practitioners

The results of this study can be used by public institutions and social organisations to improve their initiatives, programmes and policies related to nature and its services. Social media, such as Twitter, can then be a good channel for communicating environmental awareness actions, activities and campaigns by policymakers and activists, such as World Wetlands Day. In addition, being aware of the topics discussed in public opinion can be useful for companies in promoting ad hoc marketing campaigns, segmented by type of user, even more so if the social network is presented, as in our case, to many small groups, each dealing with a different aspect of ecosystem services. 

Finally, our results highlight the importance of the role of research and professional training in guiding public decision making on these issues and of complete and correct information to civil society, which is often easily influenced by the media on alarmist topics about the environment and its services as a subject [73,74].

## Figures and Tables

**Figure 1 ijerph-19-15012-f001:**
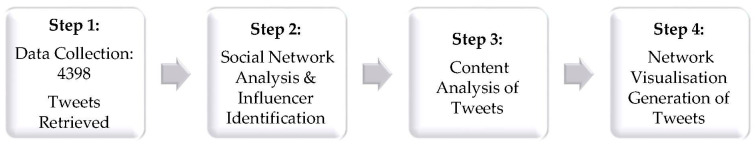
Summary of research methods.

**Figure 2 ijerph-19-15012-f002:**
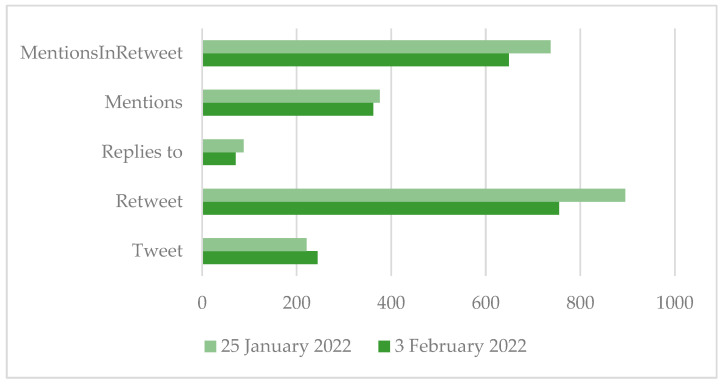
Number of tweets per category.

**Figure 3 ijerph-19-15012-f003:**
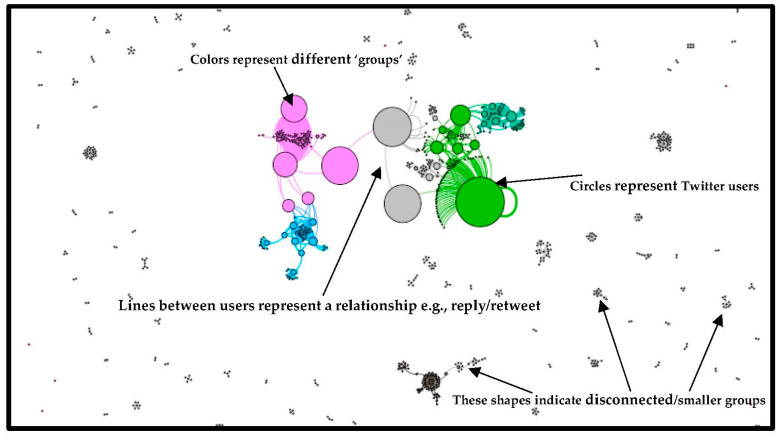
Network visual of 25 January 2022.

**Figure 4 ijerph-19-15012-f004:**
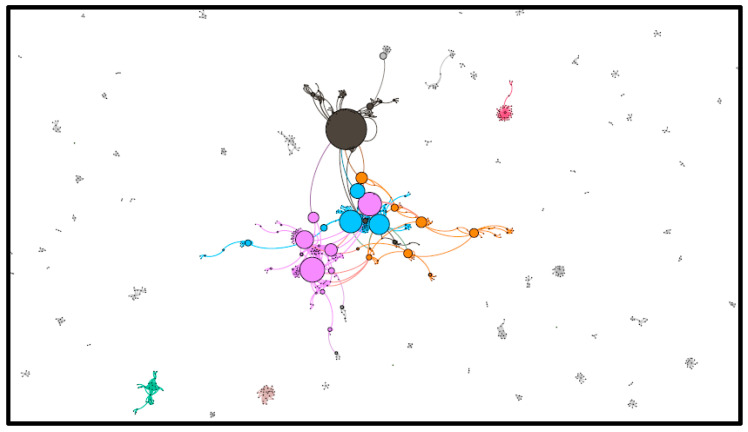
Network visual of 3 February 2022.

**Figure 5 ijerph-19-15012-f005:**
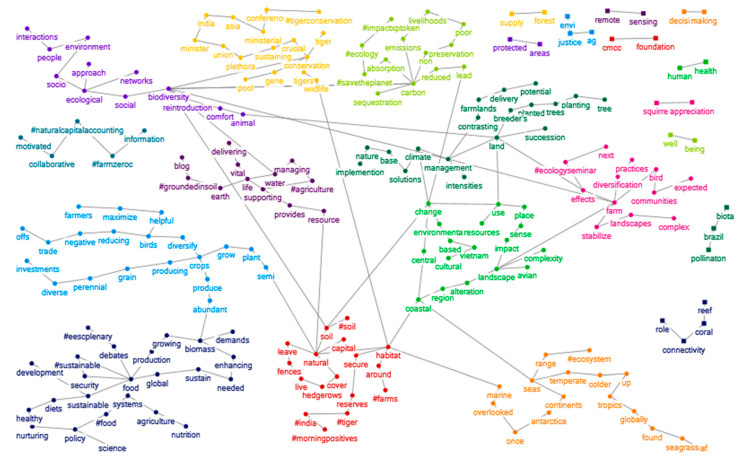
Semantic networks on tweets about ecosystem services (25 January 2022). The different colours represent the different clusters.

**Figure 6 ijerph-19-15012-f006:**
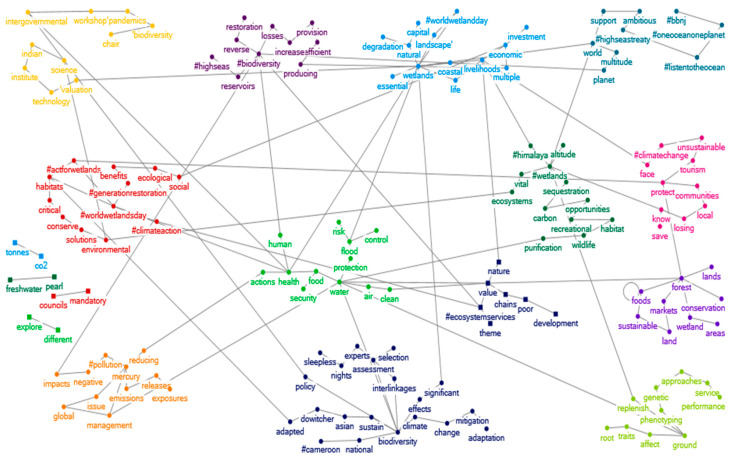
Semantic networks on tweets about ecosystem services (3 February 2022). The different colours represent the different clusters.

**Table 1 ijerph-19-15012-t001:** Macroscopic characteristics of the network.

Metrics	25 January 2022	3 February 2022
Twitter users	1427	1359
Unique relationships	2034	1905
Relationships with duplicates	283	176
Total relationships	2317	2081
Self-loops	227	246
Reciprocated Twitter users pair ratio	0.030	0.02
Reciprocated relationships ratio	0.058	0.05
Isolated Twitter users	62	99
Diameter	14	11
Average shortest path	5.7	4.4
Density	0.001	0.001
Modularity	0.82	0.84

**Table 2 ijerph-19-15012-t002:** Influential users ranked by their betweenness centrality score (25 January 2022).

Rank	Twitter User	Betweenness Centrality	Followers (n)	Network Group
1	Academic	89,525	5463	2
2	Academic	70,096	1437	25
3	Citizen	67,320	129	25
4	Academic	66,980	4914	1
5	Academic	46,670	1877	1
6	Scientific journal	43,105	23,481	1
7	Writer	34,974	1590	2
8	Non-profit organization	20,798	2701	2
9	Academic	20,033	1293	1
10	Academic	20,033	586	1

**Table 3 ijerph-19-15012-t003:** Influential users ranked by their betweenness centrality score (3 February 2022).

Rank	Twitter User	Betweenness Centrality	Followers (n)	Network Group
1	Global network	46,726	6142	4
2	Trust fund	28,226	81,907	1
3	Activist	26,748	41	1
4	Intergovernmental treaty	25,569	25,784	3
5	Non-governmental and non-profit organization	22,942	187,392	3
6	Government	20,179	179,310	1
7	Activist	16,504	11	3
8	Intergovernmental programme	13,862	1,728,382	1
9	Non-governmental and non-profit organization	12,730	2652	5
10	Scholar	11,833	385	1

**Table 4 ijerph-19-15012-t004:** Top ten hashtags on tweet about ecosystem services.

25 January 2022	Occurrence	3 February 2022	Occurrence
ecosystemservices	158	worldwetlandsday	193
ecosystem	59	ecosystemservices	174
india	59	wetlands	125
tiger	58	biodiversity	120
agriculture	56	actforwetlands	75
morningpositives	54	generationrestoration	56
biodiversity	49	ecosystem	50
soil	45	worldwetlandday	39
groundedinsoil	45	climatechange	29
climatechange	44	nature	26

**Table 5 ijerph-19-15012-t005:** Word-pairs frequencies.

25 January 2022	3 February 2022	Classes
4053	5023	0–2
1711	2143	3–10
151	302	11–30
31	28	31–50
19	10	51–100
2	1	101–1500

**Table 6 ijerph-19-15012-t006:** Most frequent word-pairs in the analysed tweets.

25 January 2022		Occurrence	3 February 2022		Occurrence
socio	ecological	103	climate	change	51
ecological	networks	95	wetlands	life	45
interactions	people	92	life	livelihoods	45
people	environment	92	livelihoods	wetlands	45
environment	socio	92	wetlands	health	45
#morningpositives	#india	54	health	#worldwetlandsday	45
#india	#tiger	54	#worldwetlandsday	#generationrestoration	45
#tiger	reserves	54	policy	biodiversity	33
reserves	secure	54	intergovernmental	science	32
secure	habitat	54	science	policy	32

#: keyword.

## Data Availability

The data presented in this study are available in Appendix A and Appendix A.

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
