# Peer review of "Ecosystem Services: A Social and Semantic Network Analysis of Public Opinion on Twitter"

_ijerph, 2022, doi:10.3390/ijerph192215012_

Round 1
Reviewer 1 Report
Overview and recommendation:
Through social and semantic network analysis, the author shows the focus of Twitter users on ecosystem services. This is an interesting study, and the results will help policy makers and ecosystem services researchers understand public thinking. However, I found that there are many problems in the article, and some of them involve the key innovation points. Since a large number of modifications may be required, I recommend rejection and resubmission. I listed the comments in detail below, hoping these could help improve the manuscript.
Major comments:
1. My biggest concern is that the two research periods selected by the author are special and may not be representative. The two periods were punctuated not only by World Wetlands Day but also by news of the expansion of a tiger reserve in India. Would public concern about ecosystem services have been significantly different in normal times without emergencies?
2. The abstract may need to be substantially revised. (1) Line 10-11. “This study analysed 4,398 tweets gathered between 25 January 2022 and 3 February 2022” The claim is inaccurate, as data were not available on January 26. Meanwhile, I would like to know why the data of January 26th was not collected. (2) Line 18-19. “Moreover, emerged the economic valuation of these services with the creation of ad hoc markets, and the focus on specific services such as the storage of atmospheric CO2 and the provision of food.” Which analysis in the paper supports this result? (2) Line 20-22. “In conclusion, the perception of social users of the role of ecosystem services can help policymakers and forest managers to outline and implement efficient forest management strategies and plans.” What aspects of the paper support this vision for forest management?
3. Line 133-135. “The maximum threshold set in the analyses was 18,000 tweet types, which probably highlights a low interest in the term ecosystem services on Twitter during the period analysed.” In the discussion section, the author writes that the software can only collect up to 18,000 tweets. So why can this conclusion be drawn here?
4. Figures 3 and 4 give very little information available. It is recommended to add a local zoom to the main section and explain what each color means.
5. All tables in the text should be in the form of a three-line table. And fonts in figures and tables should be consistent with the text.
Minor comments:
6. Line 16 “pro tection” should be “protection”
7. Line 54-55. “This figure is growing exponentially every year, at a faster rate than with the internet,”
What does faster than the internet mean?
8. Figure1. Why are the four steps represented by circles? Is it possible to go from step 4 to step 1?
9. Please modify the first row of Table 1.
10. Section 3.2.2. Why are only two days of data analyzed, not the average of the two time periods?
11. I don't use Twitter, I would like to know how the user classification in Table 2 is obtained, is it officially provided?
12. Line 181-190. Please use the correct word spacing. For example,“worldwetlandsday” should be “world wetlands day”
13. What does “#” in Figures 5 and 6 mean?
14. The conclusion is not summative and needs to be further expanded.
15. Line 373-374. Which analysis in the paper supports this claim?
16. Please unify the reference format. For example, some journal names are abbreviated, some are not, some provide DOI, some do not, and some have unknown characters.
Author Response
Major comments:
Q1. My biggest concern is that the two research periods selected by the author are special and may not be representative. The two periods were punctuated not only by World Wetlands Day but also by news of the expansion of a tiger reserve in India. Would public concern about ecosystem services have been significantly different in normal times without emergencies?
A1. We thank the reviewer for this reflection, it is the same concern we had at the beginning of the research and that is why we decided to carry out the analysis in two different time periods. Unfortunately, it has become virtually impossible to find times during the year when nothing is happening globally in relation to nature, ecosystem services and climate. There are recurring festivals, celebrations and anniversaries, socio-political changes (e.g., political elections) and society pays increasing attention to natural capital and its services. By carrying out two analyses, we were able to extract information that is common to both periods and therefore found in normal times, such as serial communicators (i.e., researchers, scientists and public institutions) and event-specific information in the time period analysed, such as World Wetlands Day and the opening of a new tiger reserve in India.
Q2. The abstract may need to be substantially revised. (1) Line 10-11. “This study analysed 4,398 tweets gathered between 25 January 2022 and 3 February 2022” The claim is inaccurate, as data were unavailable on January 26. Meanwhile, I would like to know why the data for January 26th was not collected. (2) Line 18-19. “Moreover, emerged the economic valuation of these services with the creation of ad hoc markets, and the focus on specific services such as the storage of atmospheric CO2 and the provision of food.” Which analysis in the paper supports this result? (3) Line 20-22. “In conclusion, the perception of social users of the role of ecosystem services can help policymakers and forest managers to outline and implement efficient forest management strategies and plans.” What aspects of the paper support this vision for forest management?
A2. We have modified parts of the abstract according to the proposed revisions. (1): we agree with the reviewer, and we have changed the data collection dates (examination of the second database - link S2 of the supplementary material - shows that 26 is included in the data collection). (2): the semantic analysis revealed from the content of the tweets the presence of keywords such as food production and carbon sequestration, emissions and absorption (Figure 5), nature value chains and forest markets (Figure 6). (3): the conclusion is based on what is stated in lines 46-48 and what is reported by Mota & Pickering (2021) and Johnson et al. (2021).
- Teles da Mota, V.; Pickering, C. Geography of Discourse about a European Natural Park: Insights from a Multilingual Analysis of Tweets. Society & Natural Resources 2021, 34, 1492–1509, doi:10.1080/08941920.2021.1971809.
- Johnson, T.F.; Kent, H.; Hill, B.M.; Dunn, G.; Dommett, L.; Penwill, N.; Francis, T.; González-Suárez, M. Classecol: Classifiers to Understand Public Opinions of Nature. Methods in Ecology and Evolution 2021, 12, 1329–1334, doi:10.1111/2041-210X.13596.
Q3. Line 133-135. “The maximum threshold set in the analyses was 18,000 tweet types, which probably highlights a low interest in the term ecosystem services on Twitter during the period analysed.” In the discussion section, the author writes that the software can only collect up to 18,000 tweets. So why can this conclusion be drawn here?
A3. We have changed the sentence in the methodology. The software used in the analysis set 18,000 types of tweets as the maximum threshold for the time period analysed. Our analysis yielded only 4,398 tweets. This output allowed us to draw the conclusion that there was little interest in the term ecosystem services on Twitter during that period because we could have obtained a much higher number of tweets.
Q4. Figures 3 and 4 give very little information available. It is recommended to add a local zoom to the main section and explain what each colour means.
A4. We thank the reviewer for this excellent suggestion, we have annotated Figure 3 in order to provide further information on how to interpret them. The colours simply represent the different groups that formed and this is now mentioned in the figure itself, which links to the textual explanation of the figure.
Q5. All tables in the text should be in the form of a three-line table. And fonts in figures and tables should be consistent with the text.
A5. We have corrected them.
Minor comments:
Q6. Line 16 “pro tection” should be “protection”
A6. We have made the necessary corrections.
Q7. Line 54-55. “This figure is growing exponentially every year, at a faster rate than with the internet,” What does faster than the internet mean?
A7. By that sentence we mean to point out that for the same number of people (e.g., 1 billion), social media took less time to reach that number of people than the Internet. Thank you for pointing out that this may not be clear, we have made a modification to mention ‘growth rate’.
Q8. Figure1. Why are the four steps represented by circles? Is it possible to go from step 4 to step 1?
A8. We agree with the reviewer. The process is linear, and we have corrected this within the manuscript.
Q9. Please modify the first row of Table 1.
A9. Thank you for this suggestion, w have implemented the change.
Q10. Section 3.2.2. Why are only two days of data analyzed, not the average of the two time periods?
A10. The data analysed concern two one-week time periods (as stated in the text of the manuscript, lines 104-108). The dates 25 January and 3 February refer to the days on which the data were collected. Thus, 25 January refers to the data collection of the week 17-25 January while the date 3 February refers to the period 26 January to 3 February.
Q11. I don't use Twitter, I would like to know how the user classification in Table 2 is obtained, is it officially provided?
A11. The classification of users does not follow a standardised classification scheme but refers to various works in the literature such as Ahmed et al. (2020; 2022) and Mills et al. (2019). The fact of having emphasised the role of the users, trying in some way to cluster the users, rather than their usernames allowed for reflection in the results and discussion.
- Ahmed, W.; Vidal-Alaball, J.; Downing, J.; Seguí, F.L. COVID-19 and the 5G Conspiracy Theory: Social Network Analysis of Twitter Data. Journal of Medical Internet Research 2020, 22, e19458, doi:10.2196/19458.
- Ahmed, W.; Seguí, F.L.; Vidal-Alaball, J.; Katz, M.S. COVID-19 and the “Film Your Hospital” Conspiracy Theory: Social Network Analysis of Twitter Data. Journal of Medical Internet Research 2020, 22, e22374, doi:10.2196/22374
- Mills, J.; Reed, M.; Skaalsveen, K.; Ingram, J. The Use of Twitter for Knowledge Ex-change on Sustainable Soil Management. Soil Use and Management 2019, 35, 195–203, doi:10.1111/sum.12485.
Q12. Line 181-190. Please use the correct word spacing. For example,“worldwetlandsday” should be “world wetlands day”
A12. We are unable to use the correct spacing because hashtags work on social media without spacing. So, all words prefixed with a hashtag have no spacing by official rule.
Q13. What does “#” in Figures 5 and 6 mean?
A13. The hashtag of the reference figures indicates a keyword. In social media, as in our case on Twitter, keywords are reported with a hashtag as a prefix.
Q14. The conclusion is not summative and needs to be further expanded.
A14. In the conclusion, a paragraph was added to clarify the research path undertaken by highlighting the main results achieved.
Q15. Line 373-374. Which analysis in the paper supports this claim?
A15. We have added references to support the claim (Jones et al. 2022; Lewandowsky 2021).
- Jones, G.; Hessburg, P. F.; Spies, T.; North, M. P.; Collins, B. M.; Finney, M. A.; Lydersen, J. Counteracting wildfire misinformation. Frontiers in Ecology and the Environment 2022, 20, 392-393, doi:10.1002/fee.2553.
- Lewandowsky, S. Climate change disinformation and how to combat it. Annual Review of Public Health 2021, 42, 1-21, doi:10.1146/annurev-publhealth-090419-102409.
Q16. Please unify the reference format. For example, some journal names are abbreviated, some are not, some provide DOI, some do not, and some have unknown characters
A16. We have changed the reference format, adding the full name of the journal, the DOI (when possible) and the pages.
Reviewer 2 Report
Social Network Analysis - SNA knows a growing interest in the last few years, probably because of using of tools and statistical ways for analyzing complex sets of data. Thereby it is possible to combine mathematical competencies and interests coming from the social sciences. Besides, the analyzed social media offer a vision of public opinion and collective attitudes. The Authors of this paper define appropriately their aims and demonstrate interesting outcomes, related to the ways of debating actual issues, creating social influences, and even building new networks. Their work sounds surely original, and interesting both for scientists focused on the content (in this case, ecosystem services), on the virtual environment (Twitter), and methods (SNA and semantic analysis). Limitations and suggestions for further research are clearly explained.
Strengths
- The capability to combine mathematical and social sciences approaches;
- The analysis of social media (e.g. the analysis related to Twitter) for providing a vision of public opinion and collective attitudes;
- Correct definition of the aims and methods;
- Interesting outcomes, related to the debate of social issues, like the social influences and the building of new networks;
- The originality of the study;
- The interest of both scientists focused on the content (in this case, ecosystem services), the virtual environment (Twitter), and methods (SNA and semantic analysis).
- The adequate explanation of limitations and suggestions for further research.
Weaknesses
The only weakness may be, perhaps, the difficulty for expert – but not specialized – scholars in appreciating this (still relatively new) field, and quali-quanti methods such as SNA and semantic analysis.
Author Response
Strengths
- The capability to combine mathematical and social sciences approaches;
- The analysis of social media (e.g. the analysis related to Twitter) for providing a vision of public opinion and collective attitudes;
- Correct definition of the aims and methods;
- Interesting outcomes, related to the debate of social issues, like the social influences and the building of new networks;
- The originality of the study;
- The interest of both scientists focused on the content (in this case, ecosystem services), the virtual environment (Twitter), and methods (SNA and semantic analysis).
- The adequate explanation of limitations and suggestions for further research.
Weaknesses
The only weakness may be, perhaps, the difficulty for expert – but not specialized – scholars in appreciating this (still relatively new) field, and quali-quanti methods such as SNA and semantic analysis.
Authors: We thank the reviewer for his thoughts. We have tried to simplify some points in the manuscript that might seem more difficult to read for a less experienced public, and we have added a paragraph in the conclusion to better clarify the main findings.
Reviewer 3 Report
This study analysed 4,398 tweets gathered between 25 January 2022 and 3 February 2022 related to ecosystem services, using the keyword and hashtag “ecosystem services”. The results reveal a loosely dense network in which information is conveyed slowly, with homogeneous, medium- sized subgroups typical of the community cluster structure. The perception of social users of the role of ecosystem services can help policymakers and forest managers to outline and implement efficient forest management strategies and plans. In summary, it is recommended to accept the paper after minor revisions.
Q1. Social media data paragraphs 2 Iine 57-63
Why choose Twitter as the research carrier? If an opinion questionnaire is conducted, please supplement the content and results of the questionnaire. And whether the nationality of the questionnaire will be distributed and the information on the region, because this can include more public opinion.
Q2. Research questions and hypothesis paragraphs 2 Iine 85-86
“Based on the empirical evidence found in the literature.”What specific literature are you based on? Please supplement reference references.
Q3. Data Analysis paragraphs 1-2 line 112-126
The research methods part of this paper explain too little for the use of SNA in this study, please further supplement.
Q4. User analysis paragraphs 1 line 166-167
“The number of followers is not an important metric for determining the influence of a user.”This theory is different from the previous perception, but it is not specified why.
Q5. User analysis paragraphs 1 line 166-167
What are an important metric for determining the influence of a user in this article? Please supplement.
Q6. Social network analysis Figure 3、4
The map specification of these two maps needs to be further strengthened. Please supplement the figure legend information.
Author Response
Q1. Social media data paragraphs 2 Iine 57-63. Why choose Twitter as the research carrier? If an opinion questionnaire is conducted, please supplement the content and results of the questionnaire. And whether the nationality of the questionnaire will be distributed and the information on the region, because this can include more public opinion.
A1. An opinion questionnaire was not conducted and administered, we relied on references in the literature (e.g., Pan & Vira, 2019) and the characteristics of the social media that authors in the literature enunciated (e.g., the reduced presence of fake news and disbelief, the seriousness of the social media, on a par with others such as LinkedIn, compared to social media such as Facebook and Instagram) [lines 57-63]. The choice to use Twitter was also given by the NodeXL plugin, as among the different social media it proposes to analyse (e.g., Flickr, Twitter, YouTube and Wikipedia), Twitter was the most suitable for our analysis.
- Pan, Y.; Vira, B. Exploring Natural Capital Using Bibliometrics and Social Media Data. Ecology and Society 2019, 24, doi:10.5751/ES-11118-240405.
Q2. Research questions and hypothesis paragraphs 2 Iine 85-86. “Based on the empirical evidence found in the literature. ”What specific literature are you based on? Please supplement reference references.
A2. We have added some references (Johnson et al. 2021; Pan & Vira 2019).
- Johnson, T.F.; Kent, H.; Hill, B.M.; Dunn, G.; Dommett, L.; Penwill, N.; Francis, T.; González-Suárez, M. Classecol: Classifiers to Understand Public Opinions of Nature. Methods in Ecology and Evolution 2021, 12, 1329–1334, doi:10.1111/2041-210X.13596.
- Pan, Y.; Vira, B. Exploring Natural Capital Using Bibliometrics and Social Media Data. Ecology and Society 2019, 24, doi:10.5751/ES-11118-240405.
Q3. Data Analysis paragraphs 1-2 line 112-126. The research methods part of this paper explain too little for the use of SNA in this study, please further supplement.
A3. We have added some more information about SNA around our method of detecting users, and we sign-post users to further information on this topic where further information on SNA can be found.
Q4. User analysis paragraphs 1 line 166-167. “The number of followers is not an important metric for determining the influence of a user. ”This theory is different from the previous perception, but it is not specified why.
A4. We have added a sentence in the main text to explain why the number of followers is not an important influencing factor.
Q5. User analysis paragraphs 1 line 166-167. What are an important metric for determining the influence of a user in this article? Please supplement.
A5. The important metric for a user's influence is betweenness centrality, since the higher it is the more influence the user has in allowing or blocking the exchange of information between one person/user and another in the network. We have added a sentence in the main text to explain the importance of the betweenness centrality metric.
Q6. Social network analysis Figure 3,4. The map specification of these two maps needs to be further strengthened. Please supplement the figure legend information.
A6. We thank the reviewer for this excellent suggestion, we have annotated Figure 3 in order to provide further information on interpretation.
Round 2
Reviewer 1 Report
This revision addresses most of my concerns. At present, I have four suggestions.
1. It may be better to merge very short paragraphs.
2. The first column in Table 1 should have a header.
3. In Table 2, Table 3, and Table 5, except for the first line, it may be better to keep the font thickness consistent.
4. It is suggested to add the first question in the last review comments to Chapter 5.1.
Author Response
Q1. It may be better to merge very short paragraphs.
A1. We have tried to merge some paragraphs.
Q2. The first column in Table 1 should have a header.
A2. We have added the header.
Q3. In Table 2, Table 3, and Table 5, except for the first line, it may be better to keep the font thickness consistent.
A3. We have corrected them.
Q4. It is suggested to add the first question in the last review comments to Chapter 5.1.
A4. We have added the previous first question in the limitations (chapter 5.1).